# *Drosophila* SLC22 Orthologs Related to OATs, OCTs, and OCTNs Regulate Development and Responsiveness to Oxidative Stress

**DOI:** 10.3390/ijms21062002

**Published:** 2020-03-15

**Authors:** Darcy C. Engelhart, Priti Azad, Suwayda Ali, Jeffry C. Granados, Gabriel G. Haddad, Sanjay K. Nigam

**Affiliations:** 1Department of Biology, University of California San Diego, San Diego, CA 92093, USA; dengelha@ucsd.edu; 2Department of Pediatrics, University of California San Diego, San Diego, CA 92093, USA; pazad@ucsd.edu (P.A.); saa023@ucsd.edu (S.A.); ghaddad@ucsd.edu (G.G.H.); 3Department of Bioengineering, University of California San Diego, San Diego, CA 92093, USA; j6granad@ucsd.edu; 4Department of Medicine, University of California San Diego, San Diego, CA 92093, USA

**Keywords:** solute carrier 22 (SLC22), Remote Sensing and Signaling Theory, interorgan communication, organic anion transporter, organic cation transporter, SLC22A15, SLC22A16, SLC22A18, kidney, Malpighian tubule

## Abstract

The SLC22 family of transporters is widely expressed, evolutionarily conserved, and plays a major role in regulating homeostasis by transporting small organic molecules such as metabolites, signaling molecules, and antioxidants. Analysis of transporters in fruit flies provides a simple yet orthologous platform to study the endogenous function of drug transporters in vivo. Evolutionary analysis of *Drosophila melanogaster* putative SLC22 orthologs reveals that, while many of the 25 SLC22 fruit fly orthologs do not fall within previously established SLC22 subclades, at least four members appear orthologous to mammalian SLC22 members (SLC22A16:CG6356, SLC22A15:CG7458, CG7442 and SLC22A18:CG3168). We functionally evaluated the role of SLC22 transporters in *Drosophila melanogaster* by knocking down 14 of these genes. Three putative SLC22 ortholog knockdowns—*CG3168*, *CG6356*, and *CG7442/SLC22A*—did not undergo eclosion and were lethal at the pupa stage, indicating the developmental importance of these genes. Additionally, knocking down four SLC22 members increased resistance to oxidative stress via paraquat testing (*CG4630: p* < 0.05, *CG6006: p* < 0.05, *CG6126: p* < 0.01 and *CG16727: p* < 0.05). Consistent with recent evidence that SLC22 is central to a Remote Sensing and Signaling Network (RSSN) involved in signaling and metabolism, these phenotypes support a key role for SLC22 in handling reactive oxygen species.

## 1. Introduction

SLC (solute carrier) proteins are the second largest family of membrane proteins in the human genome after G protein-coupled receptors (GPCRs) and are relatively understudied given how much of the genome they represent. SLC22 has been identified as a central hub of coexpression with almost every other SLC family and appears to be one of the major hubs of coexpression amongst SLCs, ATP-binding cassette proteins (ABCs), and drug-metabolizing enzymes (DMEs) as well as the predominant hub for coexpression with phase I and phase II DMEs [1,2]. This central position within coexpression analyses of healthy, non-drug-treated tissues highlights the crucial role that these transporters likely play in endogenous physiology, as proposed in the Remote Sensing and Signaling Theory [3]. The Remote Sensing and Signaling Theory proposes that transporters and enzymes expressed in several organs function together to maintain homeostasis via inter-organ and intra-organ communication through movement of small molecules. To better understand the systemic functionality of the SLC22 family using a highly conserved but simpler model organism than mice, we chose to disrupt this central metabolic hub in *Drosophila*. Our observation of both developmental and oxidative stress phenotypes further underscores the importance of these transporters as developmental regulators and mediators of exogenous stressors.

We utilized *Drosophila melanogaster* as a model system to gain insight into the potential physiological reasons for evolutionary conservation of SLC22 and to investigate their role in mediation of oxidative stress. Evolutionary studies suggest that, in addition to animals, the SLC22 family is conserved in members of the fungi kingdom, such as the unicellular eukaryote *S. cerevisiae*, as well as *A. thaliana* of the plant kingdom. However, these species lack physiologically “parallel” systems that could provide insight into the function of human SLC22 transporters [4]. Due to the similarities between *Drosophila melanogaster* physiology and human systems, such as shared functions of the *Drosophila* hindgut and Malpighian tubules and the human intestines and kidneys, the fruit fly serves as a valuable model for human renal and intestinal disease states. Approximately 65% of human disease-associated genes have putative orthologs in *Drosophila* and within functional regions, these fly genes can share up to 90% amino acid or DNA sequence identity with their human orthologs [5]. With identification of 25 SLC22 proteins in the *Drosophila* genome and the availability of reliable RNAi lines for many of these genes from the Bloomington Drosophila Stock Center (BDSC), the fruit fly provides a feasible platform for our overall developmental and physiological inquiry [5,6,7].

Although there are no established SLC22 orthologs between fly and human, there is evidence that some SLC22 fly genes share substrates and possibly functionality with human SLC22 members. Two of these genes, *CarT*/carcinine transporter (*CG9317*) and *BalaT*/β-alanine transporter (*CG3790*), play major roles in histamine recycling. In *Drosophila* photoreceptor neurons, *CarT* mediates the uptake of carcinine, an inactive metabolite that results from the conjugation of β-alanine and histamine [8]. Carcinine has been detected in mammalian tissues such as the human intestine and is transported by human OCT2 (SLC22A2) in both in vitro and in vivo studies [8,9]. *BalaT* mediates the recycling of β-alanine, which is necessary for histamine homeostasis in *Drosophila* photoreceptor synapses [10]. In addition to the imperative role of histamine in *Drosophila* neurotransmission, histamine and histamine receptors (HRs) have broad physiological and regulatory functionality in both the cardiovascular and central nervous systems. OCT2 and OCT3 (SLC22A3) which share high homology with CarT and BalaT, are expressed in higher order species, such as mice and humans [8,10,11,12,13]. This relationship is supported by shared substrate specificity for monoamines, such as carcinine and other neurotransmitters, as well as similar neuronal expression patterns of fly and human genes [9,14,15]. Despite *CarT* knockdowns in flies resulting in blindness and complete loss of photoreceptor transmission and *BalaT* knockdowns in flies severely disrupting vision and inhibiting photoreceptor synaptic transmission, both *Oct2* and *Oct3* knockout mice show no phenotypic abnormalities [8,10,16,17]. To better utilize *Drosophila* as a model of human SLC22 proteins and direct future studies, a homology-based analysis was performed with all fruit fly putative SLC22 orthologs in the frame of well-established SLC22 members from human, mouse, and other common model organisms. This analysis found at least 10 of the putative fruit fly orthologs within the previously defined SLC22 subclades [18]. Four of these fly genes share common ancestry with single SLC22 members which is characteristic of a direct, functional ortholog.

SLC22 members in mice, such as OAT1 (SLC22A6), OCT1 (SLC22A1), and OAT2 (SLC22A7), are transiently expressed throughout development in tissues that show minimal or no expression in adulthood [19]. *Oct1-3* (*Slc22a1-3*), *Octn1* (*Slc22a4*), *Oat1*, *Oat3* and *Rst* (*Slc22a12*) knockout (KO) mice are fertile, viable and show no general phenotypic abnormalities except for the *Oat3* KO’s decreased blood pressure, serum metabolite changes, and the *Octn1* KO’s increased susceptibility to intestinal inflammation [6,16,17,20,21,22,23,24,25]. The only SLC22 knockout mouse line with a reported clear developmental phenotype is the *Octn2* (*Slc22a5*) KO [24]. OCTN2 (SLC22A5) is the main transporter of carnitine in the bodies of both humans and mice and mutations in this gene are associated with systemic carnitine deficiency [26,27]. The *Octn2* KO line is also referred to as the JVS (juvenile visceral steatosis) line because of the defects in fatty acid oxidation due to carnitine deficiency that results in the abnormal accumulation of lipids. Without carnitine supplementation, *Octn2* KO mice develop dilated cardiomyopathy, fatty livers and steatosis of other organs, and expire in 3–4 weeks [24]. Although *Oat* KO’s (including *Slc22a12*) and *Oct* KO’s have abnormal levels of metabolites and signaling molecules, with only one clear developmental phenotype observed thus far in mice, determining the functional importance of these genes in *Drosophila* could provide insight for orthologous developmental roles in mice and humans given their interesting developmental expression patterns [19,28,29,30,31]. As an initial developmental screen, we created ubiquitous RNAi knockdowns driven by a ubiquitous *da-GAL4* driver of 14 of the putative SLC22 orthologs and observed their development. A ubiquitous driver was chosen due to the diverse expression patterns of human SLC22 members [32]. Fruit flies have distinct, easily observable developmental stages of which the egg and pupa stage are the most sensitive to environmental stressors and RNAi knockdowns [33,34]. We show that, of the 14 RNAi knockdowns, three are lethal at the pupa stage, for the first time implicating *Slc22a15*, *Slc22a16*, and *Slc22a18* genes in development. 14 out of the 24 putative SLC22 orthologs were readily available from BDSC.

Paraquat (PQ) resistance tests were performed on ubiquitously expressing knock-down SLC22 lines that progressed to the adult stage. Paraquat is an herbicide and neurotoxicant that is known to cause Parkinson’s disease [35]. Low levels of this herbicide can induce redox cycling that yields high levels of reactive oxygen species (ROS), causing systemic oxidative stress [36]. Because of this, it is used as a tool for investigation of acquired resistance to oxidative stress in *Drosophila melanogaster* [37]. As a major contributor to the pathogenesis of a multitude of human diseases, such as cardiovascular disease, metabolic syndrome, neurological disorders, and general cell and tissue degradation associated with aging, oxidative stress, and the mechanisms with which we manage free radicals, are of extreme interest [38]. Ubiquitous RNAi knockdowns of at least some SLC22 members might be predicted to affect resistance to oxidative stress because many SLC22 members transport or affect serum levels of antioxidants. Some examples observed in both mice and humans are OCTN1 (SLC22A4) and ergothioneine (EGT), URAT1 (SLC22A12) and uric acid, and OAT1/OAT3 and uric acid, dietary flavonoids, as well as TCA (tricarboxylic acid) intermediates such as the oxoacid, α-ketoglutarate [6,22,39,40]. Additionally, carcinine, the characteristic substrate of the fly SLC22 member, CarT, is transported by hOCT2 and has antioxidant properties [41]. Strikingly, our studies revealed that ubiquitous RNAi knockdown of four SLC22 genes resulted in significantly increased oxidative stress resistance at one or more time points.

## 2. Results

### 2.1. Drosophila Melanogaster SLC22 Phylogenetic and Genomic Analysis

As in mammalian genomes, fly SLC22 genes exist in clusters. The majority of SLC22 genes in *Drosophila* are found on chromosome 3R with many members found in tandem with other putative SLC22 orthologs. One notably large cluster consists of 6 SLC22 genes (*CG7333*, *CG7342*, *CG17751*, *CG17752*, *CG16727*, and *CG6231*) (Table 1). Inclusion of *D. melanogaster* orthologous genes in a homology-based analysis of all SLC22 members across a multitude of species (Table 2; Figure 1) resulted in the observation of at least four members that appear orthologous to mammalian SLC22 members (CG6356: Slc22a16, CG7458, SLC22A/CG7442:Slc22a15 and CG3168:Slc22a18) and an additional six members that can be preliminarily assigned to the individual subclades. The subclades of SLC22 are based on phylogenetic relatedness and functional characterization. They exist within two major clades—OAT and OCT. Although recently revised [42] the original definitions still stand here: Oat, Oat-like, Oat-related, Oct, Octn, and Oct/Octn-related [17]. When a GUIDANCE 2.0 alignment was performed and all sequences with a GUIDANCE score of <0.6 were removed, the only topology change observed was the omission of all SLC22A18 sequences and the reassignment of CG3168 to the large fly SLC22 transporter group, indicating that it may have sequence homology with SLC22A18, but not the other members of the Oat-related subclade. Interestingly, *CG3168* is the only putative SLC22 ortholog that is localized to the X chromosome in flies. CG6006 and CG8654 fall within the Oat-related subclade and CG6126, CG8654, Orct/CG6331, and Orct2/CG13610 appear to be part of the Oct subclade. The remaining 15 orthologs form their own group outside of the SLC22 subclades and are considered to be mostly organic cation transporters [5,18]. In summary, in flies, SLC22 appears to have at least some orthologous genes that, based off of sequence analysis, may prove to be useful models for their relatively understudied human counterparts.

### 2.2. Developmental Phenotypes of D. melanogaster SLC22 Knockdowns

One of the many advantages of using *Drosophila melanogaster* as a model organism is its distinct, easily visualized developmental stages. Additionally, RNAi knockdowns of any gene in *Drosophila* show pupal lethality at a rate of about 15% [34]. These developmental observations provide valuable information regarding the developmental function of orthologous genes that may have compensatory mechanisms in higher-order species. Three SLC22 ubiquitous knockdowns (*CG7442/SLC22A*, *CG3168*, and *CG6356*) proved to be lethal at the pupa stage when crossed with the ubiquitous *da-GAL4* driver line. Crosses were repeated three times to confirm phenotypes. CG6356 appears to be a direct ortholog of SLC22A16, a carnitine transporter related to OCTNs. In addition, *CG3168*, which also arrests at the pupa stage is a putative ortholog of the poorly understood SLC22 member, SLC22A18. To our knowledge, the murine knockouts of these genes have not been reported.

### 2.3. PQ Resistance Test of D. melanogaster SLC22 Knockdowns 

Paraquat testing is commonly used in *Drosophila* to determine oxidative stress resistance in which increased survival is correlated to increased resistance to oxidative stress [36,37]. Previous studies have established reliable dose-response curves for paraquat testing in *D. melanogaster* [45]. SLC22 transport proteins in the proximal tubule cells of the kidney take small molecules, such as the antioxidants and SLC22 characteristic substrates uric acid and ergothioneine, into cells to be later excreted [39,40,46,47]. By blocking this route of excretion, levels of these small molecules, such as antioxidants (including dietary flavonoids), are expected to increase in the *Drosophila* hemolymph and increased hemolymph levels of antioxidants would confer resistance to oxidative stress. Through paraquat testing, we show that knocking down SLC22 members in *Drosophila* significantly increases resistance to oxidative stress at different time points in at least four knock-down lines (*CG4630: p* < 0.05, *CG6006: p* < 0.05, *CG6126: p* < 0.01 and *CG16727: p* < 0.05) when compared to parent and *da-GAL4* control lines (Figure 2, Appendix A–S4). The most apparent oxidative stress resistant phenotype is observed for the knock-down of *CG6126*, showing statistically significant increased survival at 36-, 48- and 60-h time points with 100% survival of the RNAi knockdown flies for all three time points – and an average of about 40% at 36 h, 20% at 48 h, and 10% at 60 h for the parent lines which were used as a control (*p* < 0.01). In mice, it is known that SLC22 transporters like OAT1, OAT3, RST, and OCTN1 directly regulate key antioxidants such as uric acid, EGT, flavonoids, and TCA intermediates [6,25,48]. Whether or not these fly transporters directly or indirectly regulate redox states will be explored in future studies. 

## 3. Discussion

Out of the seven transporters chosen by the International Transporter Consortium and the FDA for evaluation during drug development, three (OAT1, OAT3, and OCT2) are members of the SLC22 family [49]. 17 SLC22 members are also identified as drug transporters by the VARIDT database [50]. In addition to their pharmacological importance, many of these transporters transport metabolites that play a role in the response to endogenous stressors such as oxidative stress induced by reactive oxygen species. Using *Drosophila melanogaster* as a model organism, we sought to better understand the role of SLC22 in response to oxidative stress and development. Due to the lack of information regarding SLC22 fruit fly orthologs, we attempted to characterize and classify them utilizing multiple sequence alignments and RNAi knockdowns. Because there are minimal developmental phenotypes (apart from *Octn2*) for single *SLC22* knockouts in mice despite developmentally interesting and highly dynamic expression patterns, developmental phenotypes observed in *Drosophila* could help further our understanding of how SLC22 contributes to development in other organisms as well [19,28,29,30,31]. Prior to our analysis, only three (BalaT, CarT and SLC22A) out of the 25 fruit fly SLC22 orthologs were functionally investigated beyond global tissue expression screens and homology studies [5,10,15,43,51,52]. 

Alignment of fly orthologs with the SLC22 family shows at least ten members that fall within the established Oct, Octn, Oct/Octn-related, or Oat-related subclades. There appear to be four putative orthologs to individual SLC22 members, three of which proved to be lethal in ubiquitous RNAi knockdowns. The *Drosophila* protein CG6356 shares distinct homology with SLC22A16 and RNAi knockdown of this gene resulted in arrest at the pupa stage. Based on what is known about SLC22A16 transport function and its membership in the Oct/Octn-related subclade, it is possible that this arrest is due to a systemic imbalance of both carnitine and choline. Previous *Drosophila* developmental studies have found that proper levels of either carnitine or choline are necessary for flies to reach eclosion [53]. Although SLC22A16 (FLIPT2/OCT6/CT2) has not yet been evaluated for the ability to transport choline, it is an established carnitine transporter [54]. SLC22A16 is homologous to two carnitine transporters of the SLC22 family, OCTN1 and OCTN2, which have been shown to transport acetylcholine and choline, respectively, in addition to acetylcarnitine and carnitine [55,56,57,58]. 

We observed that ubiquitous knockdown of *SLC22A/CG7442* caused arrest at the pupa stage, confirming observations made in previous studies [43]. The fly protein SLC22A/CG7442, which shares homology with human SLC22A15, has been confirmed as a transporter of characteristic OCT and OCTN metabolites MPP+, dopamine, serotonin, carnitine, TEA, choline, and acetylcholine [43]. The fly protein CG7458 also groups with SLC22A15 but lacks any phenotypic data to infer function. With further analysis, these associations could provide a basis for investigation of the endogenous function of the orphan transporter SLC22A15. Developmental tissue expression studies show transiently high expression of SLC22A15 in vital organs such as the heart, liver, and kidneys [59]. This transporter is also known to be highly expressed in white blood cells in humans, which are present at the highest concentration at birth and decrease to normal, adult levels by two years of age [55]. In combination with the observed *SLC22A/CG7442* developmental phenotype, it appears likely that putative CG7442 orthologs (such as SLC22A15) in other species may play a developmental role. 

CG316*8* groups with the Oat-related subclade, appearing to share direct ancestry with the orphan transporter, SLC22A18. Previous studies have observed high levels of CG3168 expression in glial cells during embryogenesis [60]. SLC22A18 has been shown to be expressed in low levels in the adult brain in the Human Protein Atlas, GTEx, and FANTOM5 RNA-seq studies [59]. It also has low expression levels in the human fetal brain [61]. Between the adult and fetal brain, there is a pattern of consistent expression of SLC22A18 in the cerebral cortex. The cerebral cortex consists of ~75% glial cells which could represent partly orthologous expression patterns between CG3168 and Slc22a18 in different species. Further investigation of CG3168 and its relationship to the orphan SLC22 member, SLC22A18, could build an understanding of how both of these genes are implicated in development. 

In addition to phylogenetic and developmental functional screens, RNAi knockdown fly lines that progressed to adulthood were examined for resistance to oxidative stress via paraquat resistance testing. Four knock-down lines (*CG4630: p* < 0.05, *CG6006: p* < 0.05, *CG6126: p* < 0.01 and *CG16727: p* < 0.05) showed significantly greater resistance to oxidative stress. CG16727 has no phenotypic or phylogenetic associations other than increased paraquat survival for crosses with two separate *da-GAL4* driver lines. However, it is specifically expressed in the Malpighian tubules, which are often considered somewhat analogous to the mammalian kidneys, where excretion of the antioxidant-acting oxoacids of the TCA cycle, uric acid, and flavonoids normally occurs via OATs [15,62,63,64]. TCA intermediates pyruvate, oxaloacetate, and α-ketoglutarate are known to mediate oxidative stress responses, due to direct interaction of their α-ketoacid structure with reactive oxygen species such as H_2_O_2_ [48,65,66]. Due to the conservation of metabolites between *Drosophila* and humans, we raise the possibility that RNAi knockdowns of potential OAT orthologs would be more resistant to PQ due to increased systemic levels of metabolites with antioxidant properties. The removal of the excretory route for these metabolites would result in increased serum levels, which would protect against oxidative stress. Further, investigation of a Malpighian tubule-specific knockdown of this gene would be necessary to assess this phenotype and hypothetical functionality. *CG4630*, *CG6006* and *CG6126* knockdowns showed similar oxidative stress resistant phenotypes when crossed with one *da-GAL4* driver line. All three of these transporters are expressed within the *Drosophila* excretory system but have a wider range of tissue expression than CG16727. Resistance to oxidative stress exhibited by these RNAi knockdown lines must be further examined by hemolymph analysis for classical SLC22 antioxidants such as urate, EGT, and the oxoacids of the TCA. Oxidative stress resistance is of particular interest in the search for SLC22 organic anion transporters in fruit flies. Our homology-based analyses show no unambiguous OAT orthologs in fruit flies. However, it is possible that some SLC22 fly genes transport organic anions but do not share enough sequence similarity for multiple sequence alignment programs to determine their functions.

Although some SLC22 RNAi knockdown lines may not show a significant phenotype, it has been shown that knocking down specific organic anion transporters in *D. melanogaster* can affect the expression patterns of other transporters with similar functionality, indicating a mechanism of sensing and signaling tied to organic anion, cation, and zwitterion transporters (OATs, OCTs, and OCTNs) [67,68]. Changes of expression levels of functionally similar transporters could provide further support for the Remote Sensing and Signaling Theory, in which drug-related proteins (e.g., drug transporters and drug metabolizing enzymes) and signaling molecules mediate inter-organ communication to maintain physiological balance [69,70]. For mammalian organs, a transporter and DME gene remote sensing and signaling network (RSSN) has recently been proposed [47].

Our findings show that the fruit fly is a useful model system to investigate understudied transporters, specifically SLC22A15, SLC22A16, and SLC22A18, as well as to gain functional insight into the SLC22 gene family as a whole. Additionally, confirmation of apparently strong phylogenetic relationships could result in viable models to better understand the functionality and developmental role of SLC22A16 and SLC22A18 through CG6356 and CG3168, respectively. While further study is necessary to understand the mechanism of oxidative stress resistance in certain RNAi knockdown lines, it will also be interesting to determine if there are increased levels of antioxidants in these lines and what those antioxidants might be. Given the substantial genetic and physiological conservation between mammals and *Drosophila*, these findings may support, in certain contexts, the use of fruit flies as a pre-clinical model organism for select SLC22 transporters, for instance, in elucidating their role in handling oxidative stress.

## 4. Materials and Methods 

### 4.1. Data Collection

SLC22 human and mouse sequences were collected manually from the NCBI protein database. Sea Urchin and *C. elegans* sequences were collected manually from EchinoBase (http://www.echinobase.org/Echinobase/) and WormBase (https://www.wormbase.org/#012-34-5), respectively [71,72]. Sequences were confirmed using the UCSC genome browser by searching within each available species on the online platform (https://genome.ucsc.edu/cgi-bin/hgGateway) [73]. The NCBI BLASTp web-based program was used to find sequences similar to those that were searched for manually [74]. BLASTp was run with default parameters using query SLC22 sequences from human or mouse. The database chosen was non-redundant protein sequence (nr), and no organisms were excluded. SLC22 fruit fly orthologs were determined from FlyBase (http://flybase.org/reports/FBgg0000667.html), and sequences were collected manually from the NCBI protein database [75]. Genomic locations of all transporters in question for fruit fly were determined from FlyBase. Drosophila tissue expression data was collected from FlyAtlas (http://flyatlas.org/atlas.cgi) [15]. 

### 4.2. Phylogenetic Analysis

Sequences for SLC22 were aligned using Clustal-Omega (Clustal-W) and MAFFT (Multiple alignment using fast Fourier transform) with default parameters via the online platform provided by the European Bioinformatics Institute (EMBL-EBI) (https://www.ebi.ac.uk/Tools/msa/clustalo/) [76,77,78]. Clustal-W and MAFFT produced similar topologies. These alignments were then visualized using The Interactive Tree of Life (http://itol.embl.de/) [79]. Topology confidence was additionally confirmed by branch length values, which are a result of the neighbor-joining method which calculates the number of amino acid changes between the organism at the end of the branch and the common ancestor from which it branched to visually display relatedness [80].

### 4.3. Drosophila Strains and Genetics

Drosophila stocks were fed on standard cornmeal-molasses-yeast diet and kept at room temperature [45]. *Gal4* and RNAi lines were obtained from the Bloomington Drosophila Stock Center (Indiana University, Bloomington, IN, USA) [81]. Ubiquitous RNAi via the *GAL4/UAS* was used to downregulate the following putative SLC22 transporters: *BalaT* (*CG3790*), *CG6231*, *CG4630*, *Orct* (*CG6331*), *Orct2* (*EP1027*, *CG13610*), *SLC22A* (*CG7442*), *CG16727*, *CG7333*, *CG8654*, *CG6126*, *CG6006*, *CG7084*, *CG3168*, *CG6356* [5,51]. Male SLC22 RNAi stocks were crossed to *da-GAL4* female virgins to produce an F1 generation with ubiquitous downregulation of the specific SLC22 transporters [82].

### 4.4. RNAi Developmental Screens and Paraquat Exposure

F1 offspring were observed from the egg stage through eclosion. Developmental phenotypes were defined as normal development of the F1 generation up until the failure to reach eclosion and surpass the pupa stage. Male F1 flies aged two to seven days after eclosion were tested for paraquat sensitivity as defined by survival. Both parent lines were tested in parallel as controls. Three replicates of 10 flies each were tested per strain. Flies were fed on a 3 mm Whatmann paper soaked with 10 mM paraquat (*N,N′-dimethyl-4,4′-bipyridinium dichloride*, Sigma) in 10% sucrose. Fresh paraquat was added daily. For the initial 60 h, the number of dead flies were recorded every 12 h. All tests were performed at room temperature. In order to avoid unnecessary stress, flies were not starved before adding paraquat. The significance of survival trends was assessed using one-way ANOVA followed by a post hoc Tukey’s *t*-test.

## Figures and Tables

**Figure 1 ijms-21-02002-f001:**
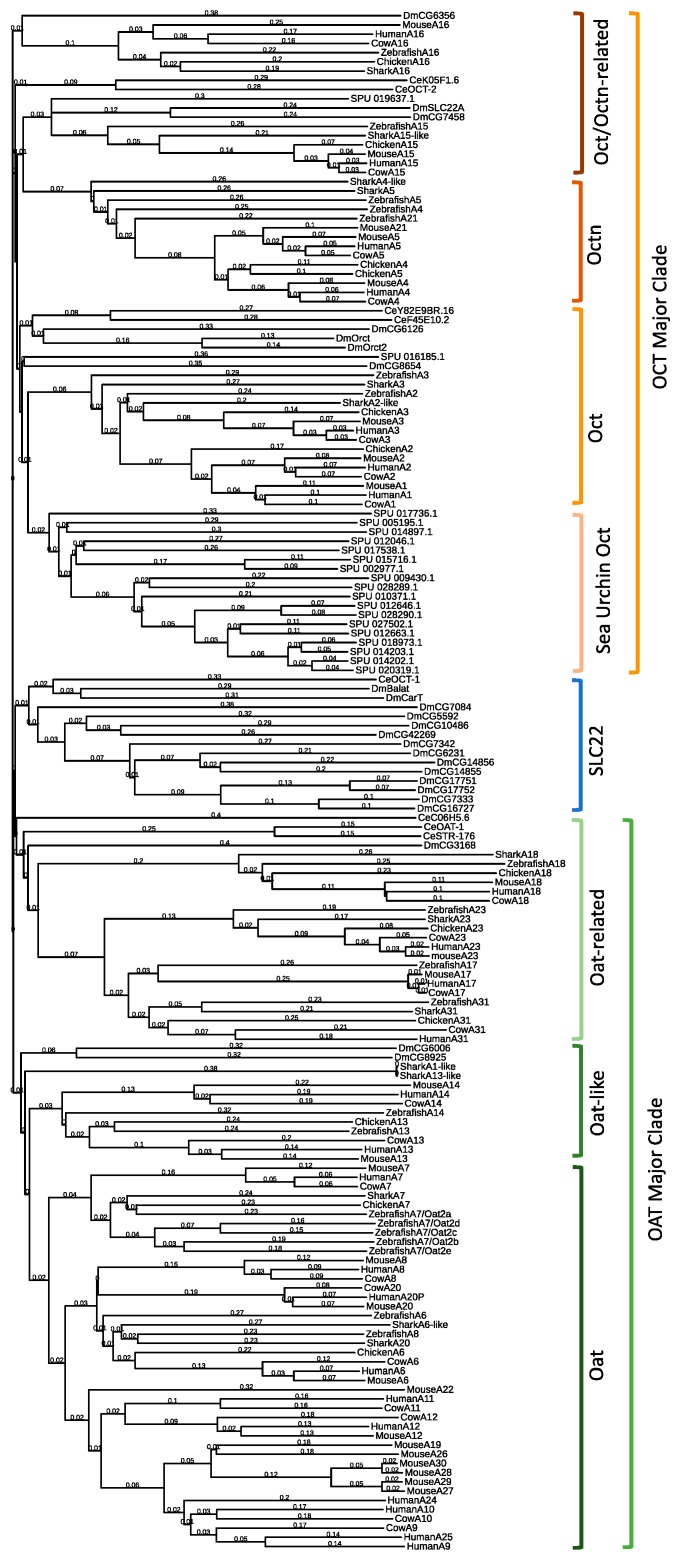
Guide tree of the SLC22 Transporter Family Using 167 Sequences. Sequences from human, mouse, cow, chicken, shark, zebrafish, sea urchin (SPU), *C. elegans* (Ce), and fruit fly (Dm) were aligned and tree was generated using Clustal Omega (using default parameters). The tree was viewed using Interactive Tree of Life (iTOL). Branch length values are calculated via the Kimura method [44]. Large sea urchin expansion within the Oct Major clade is labeled “Sea Urchin Oct”. Sequences that fall between the Oat Major Clade (green) and Oct Major Clade (orange) are denoted as SLC22 (blue).

**Figure 2 ijms-21-02002-f002:**
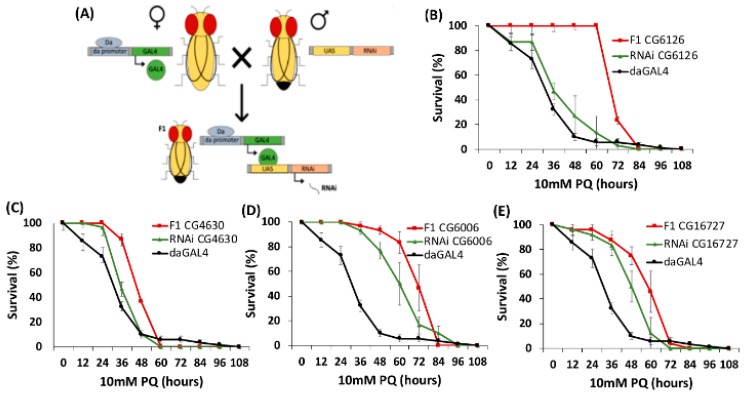
Four RNAi knockdown lines show resistance to oxidative stress. All tested lines were observed for 108 h. For each line, there is at least one time point in which the RNAi knockdown flies survived at statistically significant higher rates than both parent lines. Further information regarding statistical significance can be found in Appendix A. (**A**) Schematic of GAL4/UAS system used to generate RNAi knockdown lines. (**B**) Survival of *CG6126* knockdowns compared to parent lines (**C**) Survival of *CG4630* knockdowns compared to parent lines. (**D**) Survival of *CG6006* knockdowns compared to parent lines. (**E**) Survival of *CG16727* knockdowns compared to parent lines.

**Table 1 ijms-21-02002-t001:** Overview of SLC22 in *Drosophila melanogaster*. The following table describes all known and putative orthologs of SLC22 in *D. melanogaster*. Genes are ordered by chromosomal location and those which appear in tandem are bolded. Tissue expression was collected from FlyAtlas [15].

Gene ID	Expression Patterns		Phenotypic Data	
	Genomic Loci	Tissue/Sexual Dimorphism	Physiological Role	Substrates
*CG3168*	X: 6,720,004-6,739,986	CNS, glial specific, ubiquitously expressed in other tissues		
*CarT/CG9317*	2L: 20,727,151-20,730,282	head, brain, eye, salivary gland	histamine recycling in photoreceptor neurons	carcinine, neurotransmitters
*BalaT/CG3790*	2R: 12,872,787-2,875,012	head, brain, eye, midgut	histamine recycling in photoreceptor neurons	Beta-alanine
*CG4630*	2R: 13,215,659-13,219,247	ubiquitously expressed, except for ovary and testis		
*CG8654*	2R: 19,979,242-19,984,895	ubiquitously expressed, except for larval and adult midgut and ovaries		
*CG5592*	3L: 5,897,638-5,899,395	Testis, males only		
*CG10486*	3L: 5,904,211-5,906,699	testis		
*CG42269*	3L: 6,066,296-6,071,545	head, hindgut		
*CG7458*	3L: 21,955,110-21,958,041	ubiquitously expressed, lower expression in larval CNS and adult brain		
*SLC22A/CG7442*	3L: 21,934,704-21,938,636	ubiquitously expressed, except for ovary and testis	memory suppressor gene	MPP, Choline, Acetylcholine, Dopamine, Histamine, Serotonin, TEA, Betaine, L-carnitine
*CG14855*	**3R: 14,796,305-14,798,474**	CNS, brain		
*CG14856*	**3R: 14,798,986-14,801,163**	hindgut, midgut, heart, higher expression in larva		
*CG6006*	**3R: 16,154,982-16,171,766**	head, brain, eye, hindgut		
*CG8925*	**3R: 16,171,991-16,180,475**	head, eye, salivary gland, midgut, hindgut		
*CG6126*	3R: 16,198,000-16,203,083	ubiquitously expressed, except for testis		
*CG7333*	**3R: 19,600,635-19,602,629**	testis		
*CG7342*	**3R: 19,603,375-19,606,475**	CNS, midgut, Malpighian tubule, hindgut, testis		
*CG17751*	**3R: 19,607,254-19,609,322**	Malpighian tubule, heart		
*CG17752*	**3R: 19,607,254-19,609,322**	Malpighian tubule		
*CG16727*	**3R: 19,613,381-19,615,784**	Malpighian tubule, testis		
*CG6231*	**3R: 19,616,024-19,632,718**	Ubiquitously expressed, lower expression in adult midgut		
*CG7084*	3R: 22,298,805-22,304,420	CNS, midgut, Malpighian tubule, hindgut		
*Orct2/CG13610*	**3R: 24,273,029-24,275,728**	ubiquitously expressed, except for Malpighian tubules		
*Orct/CG6331*	**3R: 24,276,260-24,278,792**	ubiquitously expressed, lower expression in ovaries		
*CG6356*	3R: 24,283,955-24,288,617	CNS, testis	putative A16 ortholog/carnitine transporter	

**Table 2 ijms-21-02002-t002:** Sequence alignment and functional analysis of SLC22 in *Drosophila Melanogaster*. ✓addresses column title.

Gene ID	Phylogenetic Relationship		RNAi BDSC Stock ID	Phenotypes		
	Subgroup/Subclade	Transporter		Tested	Pupa Stage Arrest	PQR
*CG3168*	Oat-related	A18	29301	✓	✓	
*CarT/CG9317*	neither Oct or Oat Major Clade					
*BalaT/CG3790*	neither Oct or Oat Major Clade		67274	✓		
*CG4630*	neither Oct or Oat Major Clade		61249	✓		✓
*CG8654*	Oct subclade		57428	✓		
*CG5592*	neither Oct or Oat Major Clade					
*CG10486*	neither Oct or Oat Major Clade					
*CG42269*	neither Oct or Oat Major Clade					
*CG7458*	Octn-related	A15				
*SLC22A/CG7442*	Octn-related	A15	35817	✓	✓ [43] confirmed in this study	
*CG14855*	neither Oct or Oat Major Clade					
*CG14856*	neither Oct or Oat Major Clade					
*CG6006*	Oat related, A16 *		55282	✓		✓
*CG8925*	Oat related, A16 *					
*CG6126*	Oct subclade		56038	✓		✓
*CG7333*	neither Oct or Oat Major Clade		57433	✓		
*CG7342*	neither Oct or Oat Major Clade					
*CG17751*	neither Oct or Oat Major Clade					
*CG17752*	neither Oct or Oat Major Clade					
*CG16727*	neither Oct or Oat Major Clade		57434	✓		✓
*CG6231*	neither Oct or Oat Major Clade		63013	✓		
*CG7084*	neither Oct or Oat Major Clade		42767	✓		
*Orct2/CG13610*	Oct subclade		57583	✓		
*Orct/CG6331*	Oct subclade		60125	✓		
*CG6356*	Octn	A16	28745	✓	✓	

Our analysis consists of a phylogenetic analysis (performed with both ClustalOmega and MAFFT alignments), observational developmental phenotypes as well as Paraquat sensitivity testing, both denoted by a check mark. Each cross that arrested at the pupa stage was repeated on three separate occasions. Paraquat tests were performed on the F1 generation of a cross between SLC22 RNAi knock-down lines (obtained from BDSC) and *da-GAL4* driver parent lines. Percent survival of F1 flies was compared to both parent lines as controls. Each test consisted of 3 replicates of 10 male flies per line. * indicates a topology seen with ClustalOmega alignment only. Genes are ordered by chromosomal location and those which appear in tandem are bolded. PQR: positive paraquat resistance phenotype.

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
