# Peer review of "Drosophila SLC22 Orthologs Related to OATs, OCTs, and OCTNs Regulate Development and Responsiveness to Oxidative Stress"

_ijms, 2020, doi:10.3390/ijms21062002_

Round 1
Reviewer 1 Report
Summary:
In the manuscript “Drosophila SLC22 orthologs related to OATs, OCTs, and OCTNs regulate development and responsiveness to oxidative stress” Engelhart et al. describe their analysis of SLC22 orthologs in Drosophila melanogaster. They found that some of the fly orthologs may be orthologous to the mammalian members of SLC22 and proceeded to knockdown some of these genes to determine their functional significance and also examine the role of certain genes in ROS resistance. This manuscript is a first step forward in better understanding the role of the SLC22 family and certainly leaves more questions than answers.
Points to address:
- Does Table 1 show an overview of all of the SLC22 orthologs? I couldn’t find CG6356 in the table even though it was one of 3 mentioned to be lethal at the pupa stage. If not all of the orthologs are shown, please make that clear in the manuscript.
- There may be some redundancy in the role of different SLC22 orthologs. It is possible that other genes were not lethal because similar family members were able to take on that role. Was this examined?
- For Figure 2, it is unclear if all of the lines were examined with PQ resistance testing and only 4 showed statistically significant results or if only 4 lines were examined. If so, what was the reason for examining these 4? If all were tested (how many?), please make that clear in the results section.
- Given that there are only 2 figures, it would greatly add to the manuscript to further examine the oxidative stress phenotype – hemolymph analysis was brought up in the discussion section.
Author Response
Points to address:
- Does Table 1 show an overview of all of the SLC22 orthologs? I couldn’t find CG6356 in the table even though it was one of 3 mentioned to be lethal at the pupa stage. If not all of the orthologs are shown, please make that clear in the manuscript.
Table 1 was updated and split into two tables. Error has been fixed.
- There may be some redundancy in the role of different SLC22 orthologs. It is possible that other genes were not lethal because similar family members were able to take on that role. Was this examined?
This was not examined due to the time it would take to create double knockdowns in addition to the lack of information concerning the function of many putative fly ADME genes.
- For Figure 2, it is unclear if all of the lines were examined with PQ resistance testing and only 4 showed statistically significant results or if only 4 lines were examined. If so, what was the reason for examining these 4? If all were tested (how many?), please make that clear in the results section.
Table 1 was updated and split into two tables. Table 2 now includes a column with the flies that were tested. 4 of the tested lines showed paraquat resistance.
- Given that there are only 2 figures, it would greatly add to the manuscript to further examine the oxidative stress phenotype – hemolymph analysis was brought up in the discussion section.
This was mostly a speculation on our part. Since we don’t know the actual endogenous substrates of nearly all the fly SLC22 transporters, it would be hard to know what to look for. In addition, we no longer have the flies in house for this testing. Furthermore, given that we only had 10 days to resubmit, we were unfortunately not able to pursue this.
Reviewer 2 Report
Engelhart et al
Drosophila SLC22 orthologs related to OATs, OCTs, and OCTNs regulate development and responsiveness to oxidative stress
The manuscript describes the use and phenotypical analysis of Drosophila melanogaster knock-down mutants in several genes putatively coding for solute carriers of the SLC22 family. The study is interesting since these transporters are of great importance in physioloy but are sadly understudied. However, the manuscript could benefit from careful edition before publication.
First of all, this referee had no access to supplemental material. Therefore, there may be some criticisms that are already answered. I apologise if that is the case. On the other hand, this referee is of the opinion that every datum important to understand the work, or of primary interest for the reader, should be placed in the main text. The authors may consider moving data from the supplemental section to the main text accordingly.
My first concern is with the repetitive use of the term "ortholog". It causes confusion because it is not clear what the authors mean with it. A good example is in lines 220-221: "Four orthologs appear orthologous to individual SLC22 members,...". To my knowledge, "ortholog" is a gene or gene product that shares sequence homology AND FUNCTION with another one from a different species. There are only three SLC22 proteins in D. melanogaster for which their transported substrate (hence its function) is known (CG9317, CG3790 and CG7442)(Table 1). Out of these, only CG7442 shares enough sequence similarity with a human transporter (Slc22a15). Unfortunately, the substrate for the latter is not known. Hence, at the very most, it could be said that there are three putative orthologs, while the remaining 21 are just, at this stage, homologous sequences classified as coding for members of the SLC22 family of transporters. I would recommend the authors to revise thoroughly and modify the text so that proper use of the terms is done and confusion avoided.
Also related to writing, this reviewer had found some difficulties following the names and identities of the genes. The readability of the text would benefit from some kind of order when talking about those genes, so that readers unfamiliar with them need not jump up and down Table 1 and the text.
Gene CG7084 is not marked in Table 1 as showing arrest at pupa stage, but arrest is mentioned on lines 168-169 for that gene. Again in those same lines (and previously in the abstract) it is said that three deletion mutants showed arrest but actually they are four if data from the text and Table 1 are combined (CG3168, CG6356, CG7084 and CG7442).
The manuscript has not a list of the 13 genes that were analysed (as stated in the abstract and several times in the manuscript) nor a description of why those 13 were chosen. In addition, if all genes were attempted but there were difficulties preventing the analysis of the full set of 24, they should be mentioned. This reviewer misses some kind of markings on Table 1 to indicate that those genes were or were not studied. Surprisingly, in the Materials and Methods section there is a paragraph listing in disarray SEVENTEEN genes as the ones analysed. It is basic that this information is provided carefully and comprehensively.
Figure 1 should be complemented with the data (as Kaplan plots) of all the mutants showing statistical differences in paraquat resistance. These are just four mutants in total and, as the authors suggest, they vary in the times at which they show resistance. In my view, those data are potentially of great interest to readers.
On the other hand, Panel c of Figure 1 has very limited interest since it shows no other information that the one already in panel b, or that it is difficult to appreciate in that first panel, anyway. In other words, it is just a repetition of results and should be avoided.
No data showing the level of silencing are provided. Expression of RNAi using the da-GAL4 system is reliable, but it is common that each individual RNAi provides a different level of silencing. Without data on the silencing success for each individual gene it is very difficult to ascertain the importance of the analysis provided later on. At very least, semi-quantitative RT-PCR data should be shown for the genes displaying a phenotype.
Finally, I would advise to trim the extension of the Discussion.
Author Response
- My first concern is with the repetitive use of the term "ortholog". It causes confusion because it is not clear what the authors mean with it. A good example is in lines 220-221: "Four orthologs appear orthologous to individual SLC22 members,...". To my knowledge, "ortholog" is a gene or gene product that shares sequence homology AND FUNCTION with another one from a different species. There are only three SLC22 proteins in D. melanogaster for which their transported substrate (hence its function) is known (CG9317, CG3790 and CG7442) (Table 1). Out of these, only CG7442 shares enough sequence similarity with a human transporter (Slc22a15). Unfortunately, the substrate for the latter is not known. Hence, at the very most, it could be said that there are three putative orthologs, while the remaining 21 are just, at this stage, homologous sequences classified as coding for members of the SLC22 family of transporters. I would recommend the authors to revise thoroughly and modify the text so that proper use of the terms is done and confusion avoided.
Any instance of the word “ortholog” when referring to either of the three genes with known function was corrected to include the word putative.
- Also related to writing, this reviewer had found some difficulties following the names and identities of the genes. The readability of the text would benefit from some kind of order when talking about those genes, so that readers unfamiliar with them need not jump up and down Table 1 and the text.
Unfortunately, a better naming convention has not been established for unknown Drosophila genes and further analysis would be necessary to confidently assign nomenclature. To clarify, fly genes are italicized. Mouse genes have only the first letter capitalized and are italicized. Human genes are in upper-case and are italicized. No proteins are italicized. Fly proteins retain the gene name but are in upper case. For both humans and mice, proteins are in upper-case.
- Gene CG7084 is not marked in Table 1 as showing arrest at pupa stage, but arrest is mentioned on lines 168-169 for that gene. Again in those same lines (and previously in the abstract) it is said that three deletion mutants showed arrest but actually they are four if data from the text and Table 1 are combined (CG3168, CG6356, CG7084 and CG7442)
Thank you for pointing this out. Table 1 was updated and split into two tables. There was an error that showed CG7084 as a lethal mutant. In the new Table 2, CG3168, CG6356, and CG7442 are now marked for pupa stage arrest. The abstract was corrected to include the correct lethal knockdowns. In line 212, CG7442 was added to the description of the lethal SLC22 knockdowns.
- The manuscript has not a list of the 13 genes that were analysed (as stated in the abstract and several times in the manuscript) nor a description of why those 13 were chosen. In addition, if all genes were attempted but there were difficulties preventing the analysis of the full set of 24, they should be mentioned. This reviewer misses some kind of markings on Table 1 to indicate that those genes were or were not studied. Surprisingly, in the Materials and Methods section there is a paragraph listing in disarray SEVENTEEN genes as the ones analysed. It is basic that this information is provided carefully and comprehensively.
We apologize: 17 and 13 were errors. All instances of 17 or 13 were replaced with 14. Table 2 now shows all of the tested lines. Justification for the testing of the selected 14 lines in included in line 108.
On the other hand, Panel c of Figure 1 has very limited interest since it shows no other information that the one already in panel b, or that it is difficult to appreciate in that first panel, anyway. In other words, it is just a repetition of results and should be
Figure 2 was changed to a 5 panel figure that includes the schematic, as well as the survival plots for all 4 lines. The bar graphs that note the significant time points for each are now included in Supplementary Figures 1-4.
- No data showing the level of silencing are provided. Expression of RNAi using the da-GAL4 system is reliable, but it is common that each individual RNAi provides a different level of silencing.
All lines come from the VALIUM20 vector, which has been shown to be more reliable than VALIUM1 or VALIUM10 vectors (https://bdsc.indiana.edu/stocks/rnai/rnai_all.html). We agree that the GAL4 system is reliable, and the (Haddad-Azad) lab group where these studies have been performed has shown repeatedly that it is highly reliable in their hands. (PMCID3464109, PMCID5850797). We cite these papers in the Methods and used the same GAL4 lines in these experiments. Unfortunately, we no longer have the flies in house for this testing. And given that we have limited time to resubmit, and we expect that getting these results would take 2-3 months, we hope that, in light of the points made above, our response is deemed reasonable and satisfactory.
Reviewer 3 Report
In this study (Englehart D. et al), the authors evaluate function of SLC22 family of transporters in Drosophila by gene knockdown in support of their role in handling reactive oxygen species.
Overall this is a well written paper that presents important insights into the functional role of SLC transporters in the Fruit Fly. SLC22 tranporters are understudied in general and this paper offers new insights. The publication includes a very thorough an well-written introduction.
- Introduction- lines 41-42: Remote Sensing and Signaling Theory is referred to and would benefit from elaboration of what this theory entails given its relevance to this paper's key findings
- Introduction - lines 43-44: Is there a reference that can be cited to support that the SLC family acts as a metabolic hub in Drosophila? (other than that it is conserved in mouse)?
- Introduction – line 72: I think the author meant to say ‘which’ instead of ‘with’
- Introduction – lines 75-77: "Despite CarT knockdowns resulting in blindness and complete loss of photoreceptor 75 transmission and BalaT knockdowns severely disrupting vision and inhibiting photoreceptor 76 synaptic transmission...." is this referring to Drosophila? – or mice or humans as in previous sentence? Just needs a bit of clarification.
- Introduction - lines 84-85: "SLC22 members such as OAT1 (Slc22a6), OCT1 (Slc22a1), OAT2 (Slc22a7) are transiently 84 expressed throughout development in tissues that show minimal or no expression in adulthood." Would benefit from stating which species this is referring to.
- Table1: PQR column looks off (may be a formatting issue with the way the pdf is created by MDPI however)
- Discussion lines 238-239: Again, are these statements made with regards to Drosophila, or based on knowledge from other species? This is a bit unclear.
- Discussion line 280: Suggest spelling out OA (since it is isn’t previously defined like the abbreviation OAT is)
- Abbreviations line 356: This section is missing several other abbreviations used (ie. DME, RSSN, etc).
Overall a very good manuscript that only requires minor revisions.
Author Response
- Introduction- lines 41-42: Remote Sensing and Signaling Theory is referred to and would benefit from elaboration of what this theory entails given its relevance to this paper's key findings
A few sentences explaining the key principle of The Remote Sensing and Signaling Theory were added beginning at line 44.
- Introduction - lines 43-44: Is there a reference that can be cited to support that the SLC family acts as a metabolic hub in Drosophila? (other than that it is conserved in mouse)?
Although this has not been addressed in Drosophila, it has been seen in humans (Rosenthal et al., 2019 - in text) and is highly conserved.
- Introduction – line 72: I think the author meant to say ‘which’ instead of ‘with’
With was replaced with “which” in line 84.
- Introduction – lines 75-77: "Despite CarT knockdowns resulting in blindness and complete loss of photoreceptor 75 transmission and BalaT knockdowns severely disrupting vision and inhibiting photoreceptor 76 synaptic transmission...." is this referring to Drosophila? – or mice or humans as in previous sentence? Just needs a bit of clarification.
“In flies” was added to lines 87 and 88 to clarify that CarT and BalaT are fly genes.
- Introduction - lines 84-85: "SLC22 members such as OAT1 (Slc22a6), OCT1 (Slc22a1), OAT2 (Slc22a7) are transiently 84 expressed throughout development in tissues that show minimal or no expression in adulthood." Would benefit from stating which species this is referring to.
“In mice” was added to line 96.
- Table1: PQR column looks off (may be a formatting issue with the way the pdf is created by MDPI however)
Table was updated and split into two. The PQR column is correctly formatted.
- Discussion lines 238-239: Again, are these statements made with regards to Drosophila, or based on knowledge from other species? This is a bit unclear.
“The fly gene” was added to line 244 and “human SLC22A15” was added to line 311-312.
- Discussion line 280: Suggest spelling out OA (since it is isn’t previously defined like the abbreviation OAT is)
OA was rewritten as “organic anion” in line 372.
- Abbreviations line 356: This section is missing several other abbreviations used (ie. DME, RSSN, etc).
The following abbreviations were added:
|
RSST RSSN DME GPCR ABC BDSC HR KO URAT1 JVS UAS RNAi ROS CarT BalaT SPU TCA |
Remote Sensing and Signaling Theory Remote Sensing and Signaling Network Drug Metabolizing Enzyme G Protein Coupled Receptor ATP-Binding Cassette Bloomington Drosophila Stock Center Histamine Receptor Knockout Uric Acid Transporter Juvenile Visceral Steatosis Upstream Activation Sequence Interfering RNA Reactive Oxygen Species Carcinine Transporter Beta Alanine Transporter Strongylocentrotus Purpuratus Tricarboxylic Acid Cycle |
- Overall a very good manuscript that only requires minor revisions.
We appreciate the comment and hope the revisions are satisfactory.
Round 2
Reviewer 2 Report
Engelhart et al.
"Drosophila SLC22...[] oxidative stress"
The authors should take care of the following issues in their text:
The pdf I received came from a version that showed revisions and changes between two earlier word processor versions. A better, final, version should be uploaded
Lines 146-147 (..., indicating that it may have sequence homology with Slc22a18, but not the other members of the Oat-related subclade (Fig. S1)) call to the figure is incorrect and not clear what they mean if they actually refer to Fig. 1
Figure 2 appears repeated: the old version on top of a new version.
Also, the new version of Figure 2 should be modified to show thicker lines in the plots, as in the previous version.
The authors should double check the text for errors like that found on line 257 ("We observe that ubiquitous..."). it should read "We observed that ubiquitous..."
Author Response
Dear Editors,
We appreciate the reviews, and we have addressed all the points raised by reviewers. The following changes have been made to the manuscript: Engelhart et. al. Drosophila SLC22 orthologs related to OATs, OCTs, and OCTNs regulate development and responsiveness to oxidative stress;
Reviewer 2:
The pdf I received came from a version that showed revisions and changes between two earlier word processor versions. A better, final, version should be uploaded
All previous changes were accepted from the previous tracked-changes version and the newest uploaded tracked-changes version should only have changes pertaining to this review.
Lines 146-147 (..., indicating that it may have sequence homology with Slc22a18, but not the other members of the Oat-related subclade (Fig. S1)) call to the figure is incorrect and not clear what they mean if they actually refer to Fig. 1
We apologize. This was left from a previous version of the paper. Reference to Fig S1 has been removed.
Figure 2 appears repeated: the old version on top of a new version.
We believe this was a result of the tracked-changes feature on microsoft word. We hope this is resolved in the latest draft.
Also, the new version of Figure 2 should be modified to show thicker lines in the plots, as in the previous version.
Figure 2 has been modified to show darker, more visible lines.
The authors should double check the text for errors like that found on line 257 ("We observe that ubiquitous..."). it should read "We observed that ubiquitous..."
Line 248 (observe → observed)
Thank you for your time,
Darcy Engelhart